# Crystal Structure-Based Exploration of Arginine-Containing Peptide Binding in the ADP-Ribosyltransferase Domain of the Type III Effector XopAI Protein

**DOI:** 10.3390/ijms20205085

**Published:** 2019-10-14

**Authors:** Jyung-Hurng Liu, Jun-Yi Yang, Duen-Wei Hsu, Yi-Hua Lai, Yun-Pei Li, Yi-Rung Tsai, Ming-Hon Hou

**Affiliations:** 1Institute of Genomics and Bioinformatics, National Chung Hsing University (NCHU), Taichung 40227, Taiwan; yunk293@gmail.com (Y.-P.L.); stave0972@gmail.com (Y.-R.T.); mhho@nchu.edu.tw (M.-H.H.); 2Department of Life Science, NCHU, Taichung 40227, Taiwan; hbm486426@gmail.com; 3Graduate Institute of Biotechnology, NCHU, Taichung 40227, Taiwan; jyang@nchu.edu.tw; 4PhD Program in Medical Biotechnology, NCHU, Taichung 40227, Taiwan; 5Graduate Institute of Biochemistry, NCHU, Taichung 40227, Taiwan; 6Department of Biotechnology, National Kaohsiung Normal University, Kaohsiung 80201, Taiwan; dwhsu@nknucc.nknu.edu.tw

**Keywords:** type III effectors, peptide-binding domain, mono-ADP-ribosyltransferase, crystal structure, molecular dynamics simulation

## Abstract

Plant pathogens secrete proteins called effectors into the cells of their host to modulate the host immune response against colonization. Effectors can either modify or arrest host target proteins to sabotage the signaling pathway, and therefore are considered potential drug targets for crop disease control. In earlier research, the *Xanthomonas* type III effector XopAI was predicted to be a member of the arginine-specific mono-ADP-ribosyltransferase family. However, the crystal structure of XopAI revealed an altered active site that is unsuitable to bind the cofactor NAD+, but with the capability to capture an arginine-containing peptide from XopAI itself. The arginine peptide consists of residues 60 through 69 of XopAI, and residue 62 (R62) is key to determining the protein–peptide interaction. The crystal structure and the molecular dynamics simulation results indicate that specific arginine recognition is mediated by hydrogen bonds provided by the backbone oxygen atoms from residues W154, T155, and T156, and a salt bridge provided by the E265 sidechain. In addition, a protruding loop of XopAI adopts dynamic conformations in response to arginine peptide binding and is probably involved in target protein recognition. These data suggest that XopAI binds to its target protein by the peptide-binding ability, and therefore, it promotes disease progression. Our findings reveal an unexpected and intriguing function of XopAI and pave the way for further investigation on the role of XopAI in pathogen invasion.

## 1. Introduction

Citrus canker is a disease affecting citrus crops; it is caused by a non-indigenous bacterial pathogen *Xanthomonas axonopodis* pv. *citri* (*Xac*) [1]. This pathogen generally causes leaf spotting and blemishing of the rind of the fruit, but more severe infections result in shoot dieback and fruit drop. As *Xac* is a major threat to citrus production worldwide, farmers and governments spend their time and millions of dollars annually on prevention and disease control. Similar to many Gram-negative bacterial pathogens, *Xac* delivers effector proteins directly into host cells via the type III secretion system during the infection process [2,3]. Some of the effectors are enzymes, whereas others are transcription factors or adaptors for protein–protein interaction. They are required for the development of disease symptoms in susceptible citrus plants and for a hypersensitive response in resistant plants [4,5]. Therefore, they are valuable to the development of specific inhibitors against plant diseases [6].

XopAI, encoded by XAC3230 in *Xac*, is a putative type III effector composed of 296 amino acids. Although it has been suggested to be a pathogenicity factor for citrus canker [7,8], the role of XopAI in the virulence of *Xanthomonas* remains to be characterized. The first 43 N-terminal residues of XopAI share a sequence similarity with effectors, XopE and XopJ [9]. Further, this N-terminal sequence contains an N-myristoylation motif, which was previously examined through experiments on the XopE and XopJ proteins of *Xanthomonas campestris* pv. *vesicatoria* [10]. Effectors with this motif target cellular membranes in their hosts.

Based on the sequence of the C-terminal region (residues 120 to 286), XopAI was previously annotated as a member of the Arg-specific mono-ADP-ribosyltransferase (mART) family [11]. Bacterial pathogens use mART to alter or inhibit the activity of their target proteins in host cells through the covalent transfer of the ADP-ribose group from the cofactor NAD+ onto the Arg residue of its target protein [12,13]. In this study, we determined the crystal structure of XopAI from *Xac*. We found that the C-terminal putative mART domain of XopAI is structurally similar to that of the type III effector, HopU1 in *Pseudomonas syringae* pv. tomato. HopU1 functions as a mART in *Arabidopsis thaliana*, and it targets several RNA-binding proteins including GRP7 [14]. However, XopAI does not seem eligible to be a mART because of the change in the critical residues in the active site. From analyses of the crystal packing, we found that XopAI uses an altered mART domain to bind its own N-terminal peptide containing a conserved Arg residue. We also conducted a series of molecular dynamics (MD) simulations to investigate the structural, dynamic, and energetic properties of this protein–peptide interaction. From structural dissections, our data uncovered an unexpected function of XopAI and provided valuable snapshots that could help clarify the role of XopAI in bacterial pathogenicity.

## 2. Results and Discussion

### 2.1. Structure of XopAI

We found that the full-length XopAI protein produces two types of crystals belonging to space groups, *P*4_3_2_1_2 and *P*4_1_2_1_2 under similar crystallization conditions. The structure of XopAI was determined by bromide multiple-wavelength anomalous diffraction (Br-MAD) [15] using *P*4_3_2_1_2 crystals, and the final model was refined to a resolution of 2.01 Å (Table 1). In the Br-MAD dataset, the asymmetric unit contains four bromide ions (Appendix A), which have hydrogen-bonding contacts with backbone nitrogen atoms or Arg sidechains of the protein (Appendix A). However, the *P*4_1_2_1_2 crystals diffract to a resolution of 1.53 Å and the structure was solved via molecular replacement. Amino acid residues 1–59 of XopAI in the *P*4_3_2_1_2 crystals were not built in the model as they lacked electron density, possibly owing to the intrinsic disorder in the N terminus. Similarly, the first 58 residues of XopAI in the *P*4_1_2_1_2 crystals were not built in the model. From sequence-based predictions (Appendix A), we identified that the first 70 residues of XopAI may be highly disordered. In the *P*4_3_2_1_2 and *P*4_1_2_1_2 crystals, one XopAI molecule was found in the asymmetric unit. The statistics of the data collection and refinement for these crystals are summarized in Table 2.

Based on the crystal structure in the P43212 crystals, the overall dimensions of XopAI are ~42 × 45 × 52 Å^3^. XopAI resembles a bent right hand (Figure 1A), and it can be dissected as a two-lobe structure with an N-terminal α-helical lobe (residues 74 to 193), and a C-terminal β-sandwich lobe (residues 194 to 296) (Figure 1B). The N-terminal lobe resembles a thumb and palm and contains five α-helices (α1–α5) (Figure 1A,C), which are packed against each other to form a stable core structure. The C-terminal lobe resembles fingers and belongs to an atypical β-sandwich fold where seven strands form two sheets. One β-sheet is formed by β-strands β3 and β5, and the other is formed by β-strands β1, β2, β4, β6, and β7 (Figure 1A,C). A central cleft located between the N- and C-lobes forms a potential active site (Figure 1B).

We found the homologous proteins of XopAI in many *Xanthomonas* species (Figure 1C and Appendix A). In addition, we found potential XopAI homologs in two *Acidovorax species* (*A. citrulli* and *A. avenae* subsp. *avenae*) and *Collimonas pratensis*. *A. citrulli* and *A. avenae* subsp. *avenae* cause diseases in a wide range of economically important plants [16], while *C. pratensis* inhibits fungal growth [17]. The multiple sequence alignment shows that residues in the C-lobe are, in general, more conserved than those in the N-lobe. The N-terminal disordered region is the least conserved region in the entire protein.

### 2.2. Structural Comparison of XopAI with mARTs

To identify the structural homologs of XopAI, we compared the coordinates of XopAI against those in the Protein Data Bank (PDB) at a 90% non-redundancy level. Among 31 entries with a structural similarity (Z score higher than 2.0), we found four proteins with a Z score higher than 15.0: *P. syringae* type III effector HopU1 (PDB code 3U0J) [18], *Serratia proteamaculans* type VI secretion ADP-ribosyltransferase effector Tre1 (PDB code 6DRH) [19], rat ecto-mART ART2.2 (PDB code 1GXY) [20], and *P. aeruginosa* exoenzyme S (PDB code 6GN8) [21] (Appendix A). The mART proteins are classified into H-Y-E, variant H-Y-E, and R-S-E classes based on the conserved residues in the active site (i.e., H-Y-E stands for His-Tyr-Glu and R-S-E stands for Arg-Ser-Glu) [22]. The aforementioned four mART proteins share only 20–30% sequence similarity with XopAI, and the experimental data confirmed that they belong to the R-S-E class mARTs. XopAI and these four proteins are superimposed with a root-mean-square deviation of 2.7 Å over 191 (for HopU1), 2.9 Å over 182 (for Tre1), 2.8 Å over 181 (for ART2.2), and 2.7 Å over 174 Cα atom positions (for ExoS), respectively (Appendix A). This structural comparison result clearly indicates that XopAI folds like an R-S-E class mART. However, it is unclear if XopAI is truly a mART.

When comparing XopAI with HopU1 (Figure 2A), we found that their core folds are similar, and they adopt a mixed α/β-fold with a characteristic β-sandwich structure. XopAI shows some similarity (21% sequence identity) to HopU1; however, we noticed a significant difference in their surface charge distribution. XopAI possesses a highly negatively charged surface at its central cleft (Figure 2B), whereas a positively charged surface is required for interacting electrostatically with the negatively charged phosphate group of NAD^+^ in mARTs. As shown in Figure 2B, the central cleft of HopU1 is the active site for ADP-ribosylation. It has a positively charged surface on its right-hand side, and a negatively charged surface on the left-hand side. This negatively charged surface is presumably for the accommodation of Arg (the target residue of ADP-ribosylation). A similar surface charge distribution at the HopU1 active site can also be found in other mARTs, except that the ExoS active site is mostly positively charged (Appendix A). Based on this finding, we propose that XopAI may not be a qualified mART, and it would exert different effects on host cells. Therefore, the sequence features that contribute to this different surface charge distribution on XopAI remain unidentified.

The R-S-E class mARTs contain three conserved sequence features at the active site [23,24] (Appendix A): (1) The arom-R motif contains an aromatic amino acid followed by an Arg ([YFL]-R-X). It is located at the β-strand β1 and contributes to NAD+ binding. (2) The ARTT (ADP-ribosyl-turn-turn) loop connects β-strands, β4 and β5, and it contains a [QE]-X-E motif. The conserved catalytic Glu residue at the third position is required for NAD+ cleavage and transferase activity. (3) The STS motif in the β-strand, β3 has a sequence pattern S-[TS]-[STQ], which stabilizes the structure of the active site through hydrogen bonds with catalytic Glu residue, and other conserved NAD+ binding residues. In addition to the three features, a less conserved aliph-R motif composed of an aliphatic amino acid prior to Arg (X-[LVI]-R) resides immediately after the active site loop and contributes to NAD+ binding. In XopAI, although the motif in the ARTT loop remains unchanged, the sequence of the arom-R motif is FTG, the sequence of aliph-R motif is VLE, and the sequence of STS motif is AAS. The change from Ser and Thr to Ala in the STS motif may affect active site integrity because the Ala sidechain fails to form hydrogen bonds. The Arg to Thr change in the arom-R motif leads to the surface being electrically neutral. Moreover, the Arg to Glu change in the aliph-R motif dramatically reverses the surface charge from positive to negative. As a result of these factors, there is a change in the surface charge distribution at the central cleft of XopAI, and possibly there is a change in its cofactor preference.

These sequence alterations are widely found in XopAI homologs (Figure 1C and Appendix A). XopAI homologs in *Xanthomonas* and *Acidovorax* carry a sequence of FTG in the arom-R motif, and VLE in the aliph-R motif. In addition, the sequence of the STS motif is either AAS or ATS. Similarly, a distantly-related XopAI homolog found in *C. pratensis* has YTG in the arom-R motif and VFE in the aliph-R motif. Although its STS motif reads SSS, which is similar to mARTs, the aforementioned sequence features still requires a negatively charged surface at the central cleft. Mapping the conservation from the multiple sequence alignment onto the XopAI surface reveals that conserved residues are clustered around the central cleft (Appendix A). Similarly, HopU1 has a high level of amino acid residue conservation in its active site. This suggests that the negatively charged cleft of XopAI is still important for protein functionality. However, the function of this central cleft is unclear.

### 2.3. The Central Cleft of XopAI Has the Ability to Bind Peptides

During our inspection of the crystal packing (Appendix A), we found that the negatively charged central cleft of XopAI is important for the formation of *P*4_3_2_1_2 and *P*4_1_2_1_2 crystals. Every cleft binds a segment of the N-terminal sequence (residues 60 to 69) from a neighboring protein. Consequently, this tandem interaction allows full-length XopAI proteins to form spiral threads and fill up the crystal. To study the influence of this N-terminal region on the crystallization of XopAI, we produced an N-terminal-truncated protein, XopAI-ΔN70 (by deleting the first 70 residues) and grew the crystals. The crystals belong to space group *P*2_1_, and they diffract with a resolution of 2.26 Å (Table 2). The structure of XopAI-ΔN70 was solved via molecular replacement. In XopAI-ΔN70 crystals, the asymmetric unit contains four copies of the protein. However, we did not observe a thread-like and tandem protein packing in XopAI-ΔN70 crystals (Appendix A). This result suggests that the intermolecular association of the central cleft and the N-terminal sequence is responsible for the observed tandem packing in the crystals of the full-length XopAI protein.

Although the *P*4_3_2_1_2 and *P*4_1_2_1_2 crystal forms contain one molecule per asymmetric unit, their crystal packing environments differ significantly owing to the interaction orientation (Figure 3A). For the protein interaction mode found in the *P*4_1_2_1_2 crystals, the central cleft binds residues 59 to 70 from an adjacent protein, and it has an interface area of 673 Å^2^. However, in the *P*4_3_2_1_2 crystals, the cleft binds residues 60 to 66 of another protein, and the interface area is 470 Å^2^. In a previous study, Lo Conte et al. reported that the interface area in binary protein complexes is ~800 Å^2^ [25]; in addition, they suggested that a typical protein–protein interface may involve 22 residues, and the minimum interface area for the stability of a protein–protein complex is ~500 Å^2^. Given that protein interaction found in the two crystal forms is mediated through a small sequence of amino acid residues and that it covers only a small interface area, we propose that this interaction belongs to a protein–peptide interaction, and the central cleft of XopAI is a peptide-binding cleft. In addition, we speculate that residues spanning from 60 to 69 in XopAI constitute a candidate peptide for the binding. Intrinsic Trp fluorescence studies using a synthetic peptide ArgP14aa (Appendix A) showed that the presence of ArgP14aa increased Trp fluorescence of XopAI-ΔN70 (Appendix A), indicating a decrease in the net polarity of the environment surrounding Trp residues within the protein. In contrast, the presence of Arg did not affect the fluorescence. Figure 3B shows that among the eight Trp residues in XopAI, W154 and W237 on the cleft surface contribute to the increase in the fluorescence intensity because they will be covered during the peptide binding. We also examined the quaternary structure of XopAI using analytical ultracentrifugation and revealed a monomeric state of the protein based on the sedimentation coefficient distribution (Appendix A). The results suggested that the protein–peptide interaction found in the crystals was not strong enough to form detectable dimers in the analytical ultracentrifugation experiment.

To find more empirical data to support our hypothesis, we surveyed the crystal structures of protein–peptide complexes in the PDB, and we estimated their binding energy (Table 3). From the available structural data, we found that when a protein domain accommodates a peptide of six to 10 residues in length, it has a binding energy ranging from −3 to −10 kcal/mol. In fact, it is not easy to derive a linear correlation between the peptide length and the binding energy owing to the diverse protein structures and interaction mechanisms. For example, the SH2 domain that recognizes phosphorylated tyrosine-containing peptides has a high binding energy (−7.0 kcal/mol for a four-residue peptide). On the contrary, the IRS domain, which is another tyrosine-phosphorylated peptide-binding domain, holds a low binding energy (−4.9 kcal/mol) for binding to a peptide with nine residues. By using an empirical force field, we predicted the binding energies for the protein complexes found in the *P*4_3_2_1_2 and *P*4_1_2_1_2 crystals to be −8.3 and −7.6 kcal/mol, respectively. These values are within the binding energy range shown in Table 3, which suggests the interaction between the central cleft and its peptide is energetically reasonable. In addition, these data allow us to propose that the peptide-binding cleft of XopAI functions similar to that of VHS or WD40 domains with respect to their binding energies and interface areas.

Despite a large difference in interaction orientation, the cleft recognizes an Arg residue of the peptide in both crystal forms similarly. In the insets of Figure 3A, the close-up views clearly show that the Arg from the peptide of another XopAI (for clarity, labeled as R62* in the figure) inserts deeply into the central cleft. R62* shows a clear electron density in both crystal forms, and it is surrounded by residues W154, Y159, R260, and E265. Moreover, it has a large surface area (89% of surface area in *P*4_1_2_1_2 crystals and 83% in *P*4_3_2_1_2 crystals), and it is the most conserved residue compared to others in the peptide (Figure 1C and Appendix A). Therefore, hereafter, we specifically name this peptide the “Arg peptide.” Further, the central cleft of XopAI is an Arg peptide-binding cleft. In higher eukaryotes, up to 50% of known interactions between proteins are indeed mediated by peptides [26]. We did not find any gene of the mART family in the bacteria of the genus *Xanthomonas* so far. However, mART members can be found in the bacteria belonging to the genera *Pseudomonas* and *Mycobacterium*. Previous studies demonstrated that two *P. syringae* effectors HopU1 and HopF2 use the mART activity towards different targets to interfere with plant immune signaling [14,27]. As *Xanthomonas* species have no mART, it is possible that they use XopAI to tackle host proteins and hinder the immunity signaling in host cells. XopAI can bind to an Arg peptide-like sequence or an Arg-containing surface patch in the target protein. Subsequently, this interaction blocks the functional surface of the target protein.

In XopAI crystals, we captured two distinct Arg peptide-binding poses. In the *P*4_1_2_1_2 crystals, the backbone of the Arg peptide adopts a “W”-shaped conformation in the central cleft (Figure 3B). The N-terminus of Arg peptide is placed at the left side of the central cleft, whereas the C-terminus protrudes out of the middle of the cleft. On the other hand, the Arg peptide backbone uses a “U”-shaped conformation in the *P*4_3_2_1_2 crystals; both N- and C-termini are located in the front of the middle of the central cleft. The loop between the β-strands, β3 and β4, is called the phosphate-nicotinamide (PN) loop because it binds nicotinamide phosphate and interacts with the target Arg in canonical mARTs [23]. We found that residues in the PN loop of XopAI account for 60% of the peptide-binding interface in the *P*4_1_2_1_2 crystals, and 42% of that in the *P*4_3_2_1_2 crystals. As the PN loop interacts intimately with the Arg peptide, its conformation changes significantly coupled to the peptide conformation (Appendix A). W237 in the PN loop is particularly noticeable as its position and conformation change drastically in the two binding poses (Figure 3B). The insets of Figure 3A show that in both binding poses, the cleft uses almost identical residues to bind R62*. Figure 3B clearly shows that the residues on the lower side of the central cleft—W154, Y159, R260, E263, and E265—maintain very similar sidechain conformations in both poses to interact with R62*. More interestingly, the coordinates of the central cleft in the XopAI-ΔN70 crystals show that these residues still have similar sidechain conformations in the absence of the Arg peptide (Appendix A). W154 and Y159 are located on the putative active site loop flanked by helices α4 and α5. In canonical mARTs, the active site loop stabilizes the catalytic Glu and the target Arg, and it binds nicotinamide ribose and adenine phosphate. In XopAI, W154 and Y159 form close non-bonded contacts with the R62* sidechain. R260, E263, and E265 are on the putative ARTT loop; in particular, E263 and E265 are part of the aforementioned [QE]-X-E motif, and E265 is assumed to be the catalytic Glu in mARTs. In XopAI crystals, we observed an attractive interaction between the sidechains of E265 and R62*. However, it is unclear if E265 is catalytic in XopAI.

Figure 3C summarizes the residue interactions between the cleft and the Arg peptide. In both crystal forms, the interaction to the Arg peptide is provided by residues from the arom-R (residues F201 to G203) and STS (A229 and S230) motifs, as well as those from the active site (W154 to Q163), PN (T232 to M240), and ARTT (R260 to E265) loops. As described earlier (Figure 3B), the active site and ARTT loops contribute to R62* recognition in both crystal forms. In the *P*4_1_2_1_2 crystals, the residues of the N- and C-termini interact with the PN loop. However, in the *P*4_3_2_1_2 crystals, the N-terminus interacts downward with the active site loop, whereas the C-terminus associates with residues from the PN loop. In both crystal forms, most interaction forces are non-bonded contacts through van der Waal’s interactions. Specific interactions through the hydrogen bond are consistently found between the R62* guanidinium group and the residues W154 and T156 from the active site loop. It is noticeable that these hydrogen bonds are established between the R62* guanidinium group and the backbone oxygen atoms from residues W154 and T156 (Figure 3A inset). Therefore, this interaction may not be disturbed by unwanted sidechain conformations of W154 and T156. The structural analysis website PDBsum [28] suggested a possible salt bridge between the sidechains of R62* and E265. However, the minimum distances between the R62* guanidinium group and the E265 sidechain atoms are 3.8 and 4.1 Å in the *P*4_1_2_1_2 and *P*4_3_2_1_2 crystals, respectively. It seems that this salt bridge is considerably weak in both crystal forms.

### 2.4. MD Simulation Study Supports the Protein–Peptide Interaction of XopAI

Although the crystal structures reveal the two peptide-binding modes of the XopAI central cleft (hereafter, referred to as *P*4_1_2_1_2 and *P*4_3_2_1_2 modes for short), we wondered if the observed interaction appears in an aqueous solution. To further investigate this protein–peptide interaction, we took crystal structures as starting structures and conducted two sets of 50-ns MD simulations for XopAI complexed with the Arg peptide. Although a small decrease in the binding strength was observed during the first 8 ns in the *P*4_3_2_1_2 mode, the binding energy between the XopAI central cleft and the peptide did not show any significant change in both courses of the MD simulation (Appendix A). The root-mean-square deviation (RMSD) value with respect to the crystal structure showed that the bound R62* in the *P*4_3_2_1_2 mode altered its conformation in the initial time period of 10 ns (Appendix A). However, post this period, it held a stable structure in the range of 1–2 Å relative to the starting X-ray structure. The bound R62* in the *P*4_1_2_1_2 mode did not change drastically in conformation through the entire simulation. However, we observed that the bound Arg peptide in the *P*4_1_2_1_2 mode slowly and slightly shifted its backbone structure between 0 to 25 ns (Appendix A). In summary, these data reveal that the peptide binding of XopAI persisted until the end of the simulation, suggesting that the observed peptide-binding modes in the crystals are sufficiently stable.

We further monitored structural evolution along the last 25 ns of the MD simulation to analyze the possible Arg peptide-binding modes in the aqueous solution. Figure 4A shows that the peptide possesses a similar backbone conformation throughout the simulation. To reflect the structural flexibility in each amino acid, we calculated the root-mean-square fluctuation (RMSF) of all Cα atoms in the peptide relative to the central cleft (Figure 4B). We found that R62* is the most stable residue in both binding modes; it has a Cα RMSF value of 0.83 Å in the *P*4_1_2_1_2 mode and that of 0.86 Å in the *P*4_3_2_1_2 mode. On the other hand, both the N- and C-termini of the peptide show high dynamics in the cleft, especially the N-terminus. When we estimated the energetic contribution of each residue to the binding energy from the MD simulation trajectories (Appendix A), we noticed that R62* plays a critical role in binding to the central cleft because it has a high contribution energy. This finding is coherent with the low Cα RMSF value of R62* seen in Figure 4B.

The binding free energy calculation based on the last 25-ns simulation shows that both binding modes have favorable van der Waals, electrostatic, and non-polar solvation free energy values (Table 4). The most favorable contributions to the binding process arise from electrostatic interactions; this finding is in agreement with the proposed roles of R62* and the negatively charged cleft. However, the concurrent polar solvation free energy contributes negatively to the binding free energy. The total binding free energy for the *P*4_1_2_1_2 mode is calculated as −368.5 kcal/mol, whereas that for the *P*4_3_2_1_2 mode is −235.1 kcal/mol. Thus, the predicted binding free energy for the *P*4_1_2_1_2 mode is higher than that for the *P*4_3_2_1_2 mode. This result is in agreement with the result calculated from the crystal structures in Table 3.

While R62* functions as the key residue responsible for the binding, residues at the R62*-binding site provide specific and intimate interactions with R62* (Figure 4C). The R62* guanidinium group has hydrogen bonds with backbone oxygen atoms from residues W154, T155, and T156. It also interacts with the E265 sidechain via a salt bridge. This explains the observed high contribution energy predicted for E265 (Appendix A). However, the hydrogen bond network in the *P*4_1_2_1_2 mode is more intense than that in the *P*4_3_2_1_2 mode. In addition, we found that the coordinates of W154 and E265 in the *P*4_3_2_1_2 mode have more fluctuations compared with those in the *P*4_1_2_1_2 mode (Figure 4D). The rising fluctuation in E265 may cause a weak interaction with the R62* guanidinium group. Similarly, a high RMSF value of W154 may result in loose non-bonded contacts between the sidechains of R62* and W154. Appendix A revealed that R260 plays a negative role in Arg peptide binding because it has a repulsive force to the incoming R62* guanidinium group. Fortunately, E263 along with D157 reside near R260, and they neutralize the positive charge from the R260 sidechain.

Figure 4E shows the time series of the hydrogen bond occurrences at the Arg-binding site. As mentioned earlier, there are slight differences in the hydrogen bond pairs when we compare structures from the crystals with those from the MD simulation. In both binding modes, the relative positions of the R62* sidechain and backbone oxygen atoms from residues W154, T155, and T156 did not change considerably. Therefore, the hydrogen bond patterns formed by the backbone oxygen atoms are quite consistent during the entire simulation. In the *P*4_1_2_1_2 mode, the backbone oxygen atoms from residues W154 and T156 form three obvious hydrogen bonds with the R62* guanidinium group (Figure 3A insets and Figure 4C). However, in the *P*4_3_2_1_2 mode, only the hydrogen bond formed between the R62 guanidinium group and the backbone oxygen atoms from residue W154 is significant. Figure 4E shows that during the transition from the crystal to a simulated solution condition, E265 revises its sidechain conformation for 10 ns and then moves closer to the R62* guanidinium group. As a result, the salt bridge between E265 and R62* appears to be durable in the course of the 50-ns simulation. Further, the hydrogen bond between W154 and E265 is closely related to their sidechain motions. The MD simulation showed that the W154-E265 hydrogen bond remained in the *P*4_1_2_1_2 mode; however, it was disrupted in the *P*4_3_2_1_2 mode (Figure 4E). Consequently, W154 and E265 in the *P*4_3_2_1_2 mode interact more firmly with R62* when compared to those in the *P*4_3_2_1_2 mode, as shown in Figure 4C,D.

### 2.5. MD Study for the PN Loop Dynamics of XopAI in Response to the Arg Peptide Binding

To analyze the local dynamics of the apo and peptide-bound states, we calculated the average RMSD among the available XopAI crystal structures (Figure 5A). The RMSD plot shows that three loops in the C-lobe of XopAI show a large displacement in response to Arg peptide binding: the PN loop, the ARTT loop, and the loop between β-strands β6 and β7. We called the third loop “PR loop” because it contains conserved Pro and Arg (Figure 1C and Appendix A). However, we were unable to infer its function from the current structural information or the knowledge of mARTs. As described earlier, residues on the PN and ARTT loop interact directly with the Arg peptide (Figure 3C). The motion of the PR loop is associated with structural change in the PN loop because they are in close contact with each other (Appendix A). Among the three loops, the residues of the PN loop have relatively high RMSD values. This finding suggests that the PN loop may be important for the target recognition of XopAI, and it encourages us to explore the influence of peptide binding on the dynamics of the PN loop.

As displayed in Figure 5B, the PN loop adopts different conformations in available crystal structures. The loop structure found in the *P*4_1_2_1_2 peptide-bound state is in a stretched conformation, whereas that in the monomer D of the ΔN70 crystal structure is in a contracted conformation. Importantly, the PN loop conformation is strongly associated with the volume of the central cleft. For example, the central cleft in the *P*4_1_2_1_2 peptide-bound state has a volume of 722 Å^3^; however, that in the monomer D of the ΔN70 crystal structure has a volume of 159 Å^3^. As W237 is located at the tip of the PN loop, and it has the highest RMSD value, we used the Cα distance between W237 and T202 (the center of the arom-R motif) to differentiate between the conformations of the PN loop. We found that W237 has distinct sidechain positions (Figure 5C): in *P*4_1_2_1_2 peptide-bound state, it a protruding sidechain, whereas in the monomer D of the ΔN70 crystal structure, it turns its sidechain inward toward the central cleft. Therefore, we chose the distance between the mass center of the W237 sidechain and the E265 Cα atom to represent the sidechain position of W237.

Based on the results of the MD simulation, we calculated the free energy landscapes for the structural transition of the PN loop in the absence and presence of the Arg peptide (Figure 5D). The structures generated by the MD simulation are projected onto two reduced dimensional coordinates—PN loop conformation and W237 sidechain position—and their free energies are estimated by their occurrence probability. Accordingly, it was found that local optimal structures reside at deep “energy wells” on the landscape. The energy landscape for apo structures shows several wells connected by valleys, suggesting that the PN loop can shift between several stable conformations freely in the absence of the Arg peptide. The well C is a big well occupying the central part of the landscape. Earlier, we mentioned that the asymmetric unit in the XopAI-ΔN70 crystals contains four identical monomers. When mapping these structures onto the energy landscape, we found that three monomers share a similar PN loop structure near well C, whereas the other one has a distinct loop structure near well E. Unlike apo structures, the peptide-bound structures constitute a discontinuous energy landscape. The PN loop structures in the *P*4_1_2_1_2 and *P*4_3_2_1_2 peptide-bound states stay in wells G and H, respectively. However, the landscape shows no obvious valley bridging the two wells. This result implies that there is a high energy barrier for the PN loop structure during the exchange between the two peptide-binding modes.

Although the intermediate structures during the course of the MD simulation could not be captured, we are still curious about whether the two peptide-binding modes can switch from one to another. We employed protein structure morphing [29] to conjecture the conformational transitions between the two binding modes, and then, we estimated the binding free energy of the transition states (Appendix A). The calculation revealed that each transition state has a considerably negative value of binding energy, which suggests that XopAI can hold the peptide well during the transition between the two binding modes. Furthermore, the predicted path of the conformational transition fits well into the valley on the energy landscape for apo structures (Appendix A). This result supports the reliability of our structural morphing study, and it paves the way to explore the importance of individual residues on the PN loop.

## 3. Materials and Methods

### 3.1. Plasmid Constructions

Full-length XopAI from *Xanthomonas axonopodis* pv. *citri* strain XW19 was amplified by PCR using forward primer 5′-ACTGCAT**ATG**GGGTTATGCACTTCAAAGCCGA-3′ (the underlined text represents the *NdeI* site; the bold text, the start codon) and reverse primer 5′-CATGGTCGAC**CTA**CGCGATCTGGCTTTGATAAATC-3′ (the underline text represents the *SalI* site; the bold text, stop codon). The PCR product was digested with *NdeI* and *SalI*, and then, it was subcloned into a pET28a vector, yielding plasmid pET28a-XopAI. In this vector system, a 6xHis tag followed by a thrombin cleavage site are attached at the N-terminal of XopAI. Similarly, the truncated form of XopAI (XopAI-ΔN70) was amplified by PCR from the plasmid pET28a-XopAI using the primers: 5′-CTGGTGCAT**ATG**AACACCAGCGATCTGATAAAGC-3′ (the underline text represents the *NdeI* site; the bold text, start codon) and 5′- TGGTGTCTCGAG**CTA**CGCGATCTGGCTTTGATAAATC-3′ (the underline text represents the *XhoI* site; the bold text, the stop codon). The PCR product was inserted into the p11x vector (a modified p11 vector with more restrictions sites at the multiple cloning site), yielding plasmid pET11x-XopAI-ΔN70. In this vector system, a 6xHis tag and a thrombin cleavage site are added at the N-terminus of XopAI-ΔN70. These constructs were verified by DNA sequencing, and then transformed into *E. coli* strain BL21(DE3)RIL^+^ competent cells.

### 3.2. Protein Expression and Purification

The expression and purification procedures for full-length and truncated XopAI proteins are similar. Transformants were cultured at 37 °C in the LB medium containing the antibiotic (kanamycin for pET28a-XopAI and ampicillin for pET11x-XopAI-ΔN70) and chloramphenicol until OD_600_ reached 0.1, and then, it shifted to 20 °C for further incubation. After OD_600_ reached 0.4, isopropyl β-D-thiogalactopyranoside was added to reach a final concentration of 0.2 mM, and then, it was incubated for a further 20 h. Bacteria were collected by centrifugation, and suspended in an ice-cold lysis buffer (10 mM imidazole, 300 mM NaCl, 0.25% Triton X-100, 10 mM β-mercaptoethanol, 1 mM PMSF, 50 mM NaH_2_PO_4_, pH 8.0). After the resuspended cells were disrupted using microfluidizer M-110EH (Microfluidics, USA), cell debris was removed by centrifugation at 17,000× *g* for 15 min, and the clear lysate was applied onto a Ni-NTA superflow column (QIAGEN). After washing with a wash buffer (20 mM imidazole, 300 mM NaCl, 50 mM NaH_2_PO_4_, pH 8.0) to remove nonspecific-binding proteins, the 6×His-tagged recombinant protein was eluted with an elution buffer (250 mM imidazole, 300 mM NaCl, 50 mM NaH_2_PO_4_, pH 8.0). The eluted protein was dialyzed against a dialysis buffer (200 mM NaCl, 10 mM β-mercaptoethanol, 20 mM Tris-HCl, pH 7.4), and concentrated using a stirred ultrafiltration cell (Millipore). Finally, the full-length XopAI with a concentration of 10 mg mL^−1^ was used for crystallization. Due to its poor solubility, XopAI-ΔN70 was concentrated to 6 mg mL^−1^ for crystallization.

### 3.3. Protein Crystallization

XopAI crystals were grown via the hanging-drop vapor-diffusion method at room temperature. The *P*4_3_2_1_2 crystals of full-length XopAI were grown by mixing 1 μL of protein solution with 1 μL of a crystallization buffer containing 15% (*v*/*v*) PEG 400, 5.5% (*w*/*v*) PEG 20000, and 50 mM KH_2_PO_4_, pH 8.0. The *P*4_1_2_1_2 crystals of full-length XopAI were grown by mixing 1 μL of protein solution with 1 μL of 10 mM β-mercaptoethanol and 1 μL of crystallization buffer. The buffer is similar to that for the *P*4_3_2_1_2 crystals, and it consists of 16.5% (*v*/*v*) PEG 400, 6.5% (*w*/*v*) PEG 20000, and 50 mM KH_2_PO_4_, pH 8.0. The crystals of XopAI-ΔN70 were grown by mixing 2 μL of protein solution and 1 μL of crystallization buffer containing 10% (*v*/*v*) 2-propanol, 20% (*w*/*v*) PEG400, and 100 mM HEPES-NaOH, pH 9.5.

### 3.4. Data Collection and Structure Determination

All crystals were flash-cooled in liquid nitrogen. For halide phasing, bromide derivatives were obtained by soaking the *P*4_3_2_1_2 crystals in a crystallization buffer supplemented with 1 M NaBr for 30 s before flash cryocooling. Data collection was carried out at beamlines 13B1, 13C1, and TPS 05A in the National Synchrotron Radiation Research Center (NSRRC), Hsinchu, Taiwan. The program iMosflm was used for data processing [30]. Phases for XopAI were initially determined by Br-MAD, and automatic model building was performed using the program package PHENIX [31]. Data statistics for the Br-MAD dataset are shown in Table 1. Four bromide ions were found when resolving the phase (Appendix A). After density modification, the overall figure of merit increased from 0.51 to 0.74. More than 50% of residues in one asymmetric unit were traced into the experimental electron density map. The remaining residues were manually built with COOT [32]. The model for the *P*4_3_2_1_2 crystal was refined using a native dataset with a resolution of 2.01 Å (Table 2). The models for the *P*4_1_2_1_2 crystal and the XopAI-ΔN70 crystals were phased via molecular replacement with phenix.automr from the PHENIX suite [31]. All refinements were performed with phenix.refine from the PHENIX suite. The final models were evaluated using PROCHECK, which showed a good stereochemistry according to the Ramachandran plot [33]. All structure figures were rendered by PyMOL [34]. The atomic coordinates and structure factors have been deposited in the Protein Data Bank, www.pdb.org (PDB ID codes 6KLY [XopAI in space group *P*4_3_2_1_2], 6K93 [XopAI in space group *P*4_1_2_1_2], and 6K94 [XopAI-dN70]).

### 3.5. Sedimentation-Velocity Analytical Ultracentrifugation

The sedimentation velocity experiments of XopAI were carried out using an analytical ultracentrifuge (Model Optima XL-A, Beckman, USA). The protein was prepared in 200 mM NaCl, 10 mM β-mercaptoethanol, 20 mM Tris-HCl, pH 7.4. The experiments were performed at 20 °C with a Beckman An-50 Ti rotor speed of 42,000 rpm. The protein samples were continuously monitored by UV absorbance at 280 nm with a time interval of 480 s and a step size of 0.002 cm. Data collected using interference optics were analyzed in terms of the size distribution functions c(s) using the software SEDFIT [35].

### 3.6. Fluorescence Spectroscopic Assay

The peptide ArgP14aa (SSSPRPLSPLVELN) was synthesized and purified by high-performance liquid chromatography to a purity of 95%, and confirmed by mass spectrometry (MDBio Inc., Taipei, Taiwan). L-arginine (purity ≥ 98%) was purchased from Sigma-Aldrich (St. Louis, MO, USA). The buffer of the XopAI-dN70 protein was changed to 50 mM NaH_2_PO_4_, pH 8.0. In the ligand-induced fluorescence-change experiment, a final concentration of 5 μM XopAI-dN70 was incubated with various concentrations of ligand at 4 °C for 30 min. The tryptophan fluorescence was measured using a fluorescence spectrophotometer (Model F-4500, Hitachi, Japan) equipped with a cuvette of a 1 cm light path. The excitation wavelength was 280 nm, and the emission data were collected between 300 and 400 nm at 4 °C. For the static measurements, all of the measurements were recorded in triplicate.

### 3.7. Bioinformatic Analyses

Sequence similarity searching with BLASTP and TBLASTN identified potential XopAI homologs in GenBank [36]. CD-HIT [37] removed redundant sequences that are 100% identical. Multiple sequence alignments were performed using MAFFT [38] and the corresponding images were generated using the web server ESPript 3.0 [39]. Natively disordered regions of XopAI were predicted using DISOPRED [40], PONDR [41], and SPOT-disorder [42]. Structure-based sequence alignment shown in Appendix A was produced by Modeller [43]. The structural similarity search was performed with the DALI Server [44]. Amino acid conservation was calculated and plotted onto the protein surface using the ConSurf server [45]. The protein–protein interaction was analyzed by PDBePISA [46] and PDBsum [28]. Figure 3C is modified from an output of PDBsum. The binding free energy of the protein–peptide interaction in available crystal structures was estimated with FoldX [47]. The size of the protein cleft was measured by the CASTp server [48].

### 3.8. MD Simulations

The general procedure is described as follows: First, the starting structure was subjected to energy minimization using GROMACS version 4.6.7 [49] with OPLS-AA force field in an implicit solvent model. The protein was then immersed in an orthorhombic water box in the GROMACS molecular dynamics simulation and the net charge was neutralized by the addition of sodium or chloride ions (at 150 mM salt concentration). Long range electrostatics were handled using the particle mesh Ewald method. The steepest descent energy minimization was used to remove possible bad contacts from the initial structures until energy convergence reached 1000 kJ/(mol·nm). The system was subject to equilibration MD at 300 K and normal pressure constant (1 bar) for 100 ps under the conditions of position restraints for heavy atoms and LINCS constraints. Finally, the production MD was performed under constant pressure and temperature at 1 bar and 300 K, respectively. The bond-angle degrees of freedom from hydrogen atoms were removed using the virtual site algorithm to allow a 2-fs time step. MD trajectories were recorded at 100 ps intervals. Structure snapshots were visualized and analyzed with PyMOL [34] and the built-in utilities of GROMACS.

For the simulation study of the protein–peptide interaction, the starting complex structures (*P*4_1_2_1_2 and *P*4_3_2_1_2 modes) were generated by PyMOL according to the crystallography symmetry operation. Each system was simulated for 50 ns. The interaction energy of the protein–peptide complex was calculated using the g_mmpbsa tool [50]. For the simulation study of the PN loop conformation, the starting structures in the apo form were obtained from the four asymmetric monomers in XopAI-ΔN70 cells, and the monomers in the *P*4_1_2_1_2 and *P*4_3_2_1_2 crystals. Each system was simulated for 50 ns, and the Gibbs free energy landscape was composed using the GROMACS utility g_sham. The energy landscape for the peptide-bound state was constructed from the two 50-ns MD trajectories of the *P*4_1_2_1_2 and *P*4_3_2_1_2 protein–peptide complexes. For the simulation study of the peptide-binding mode transition, the interpolated trajectory between the two binding modes was created using the Yale Morph2 Server [51]. Each structure of the transition states was solvated in a water box, and it was subject to a 100-ps MD simulation with position restraints on the Cα atoms to maintain the specified conformation. The corresponding binding free energy was calculated from 10 snapshots of the resulting trajectory.

## 4. Conclusions

While the primary source of information is experiments, the functions of many proteins are computationally annotated by sequence-based similarity search. However, there is accumulating data showing that members of an enzyme superfamily exhibit diversity with respect to their substrate preference and the reactions they catalyze [52,53]. These variations stem from incremental mutations on the catalytic, substrate-binding, and allosteric sites. Understanding the molecular basis of these functional variations is important for the accuracy of genome analysis, and it is beneficial for protein design. The structural data presented here revealed that XopAI possesses a peptide-binding domain with a sequence and structural similarity to that of the members of the mART family. However, its cofactor-binding ability seems impaired or altered owing to amino acid substitutions in the putative active site. Owing to a lack of experimental validation, we cannot conclude whether XopAI is also an enzyme.

The discovery of new pathogen effectors and the characterization of their activities provide new insights into how pathogens remodel host cells for their own benefit. The studies of effectors also offer opportunities for the development of tools to explore host cell biology in the absence of disease. In this study, we presented the crystal structure of XopAI; it revealed that XopAI shares structural similarity to mARTs; however, it may have a distinct function from that of mARTs. The structural analysis and the MD simulation study uncovered the molecular basis of interaction between the XopAI peptide-binding domain and the Arg peptide. Furthermore, these data suggested that XopAI evolved to bind an Arg peptide-like sequence or an Arg-containing surface patch in its target protein, which is important for disease progression or plant immunity. The next challenge is to identify plant target protein(s) of XopAI, which are crucial to elucidate its molecular function.

## Figures and Tables

**Figure 1 ijms-20-05085-f001:**
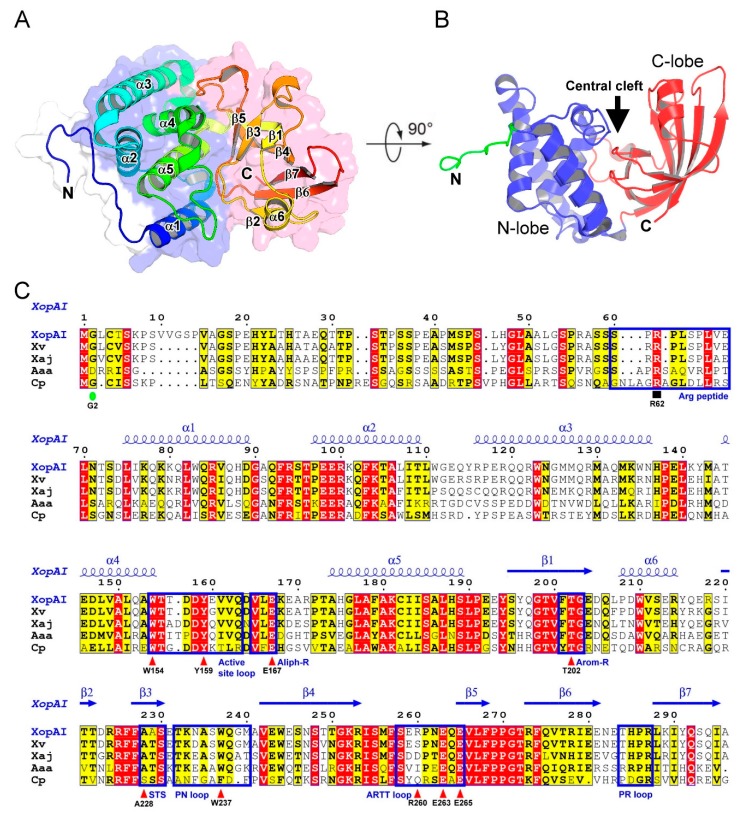
Crystal structure of XopAI and protein alignment with its homologs. (**A**) The ribbon model that shows the structure of XopAI spanning residues 60 to 296. It is colored in the rainbow scheme (from N- to C-terminus, from blue to red), and its secondary structural elements are labeled. Regions that belong to the N- and C-lobes are contoured in blue and red shadows, respectively, while the N-terminal disordered region is in white. (**B**) A view rotated by 90° relative to (A) showing the central cleft formed between the N- and C-lobes (blue and red ribbons, respectively). The N-terminal disordered region is highlighted in green. The central cleft is indicated by an arrow. (**C**) The sequence alignment of XopAI homologs; we selected five representative sequences for this concise alignment, and a more comprehensive version with 17 sequences is shown in Appendix A. The conserved residues are shaded yellow and identical residues are shaded red. Secondary structure elements based on XopAI crystal structure determined in this study are displayed above the alignment. Blue boxes outline those important regions in XopAI. Key residues in the central cleft are marked with red triangles and labeled according to XopAI sequence. A green oval shows the putative myristoylation site, and a black square indicates the crucial Arg in the Arg peptide. The following bacteria strains were analyzed: XopAI (*Xanthomonas axonopodis* pv. *citri*, GenBank accession no.: WP_011052119, this study), Xv (*X. vesicatoria* strain LM159, CP018470), Xaj (*X. arboricola* pv. *juglandis* strain Xaj 417, CP012251), Aaa (*A. avenae* subsp. *avenae*, AVS84630), and Cp (*Collimonas pratensis*, WP_061944107).

**Figure 2 ijms-20-05085-f002:**
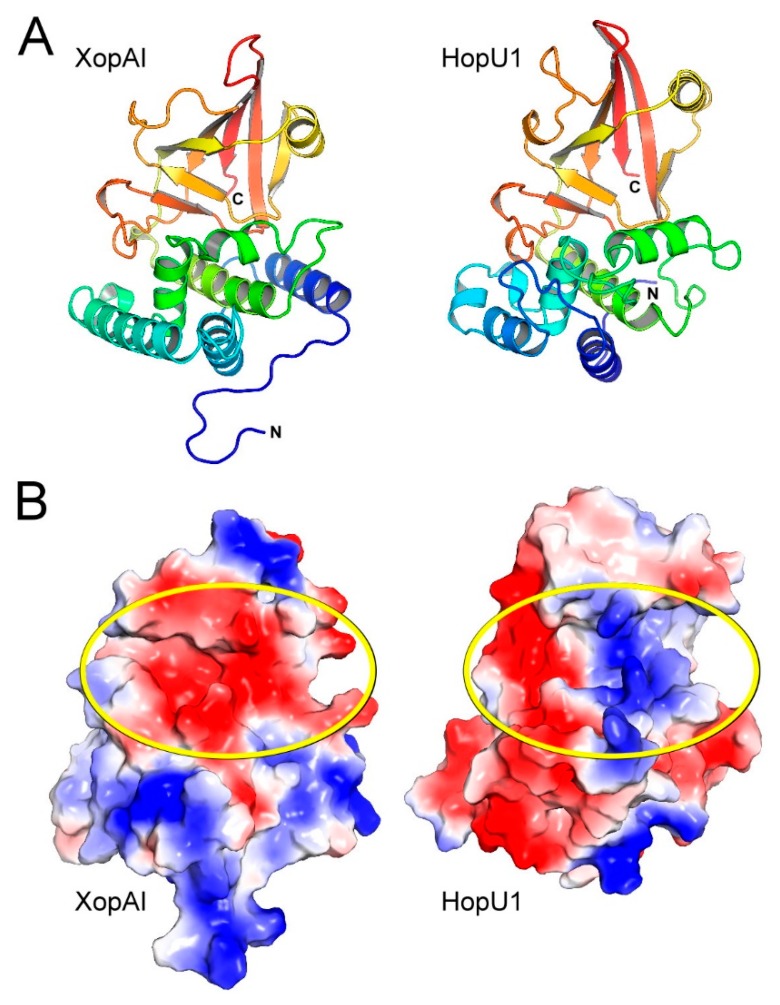
Structural comparison between XopAI and HopU1. (**A**) Ribbon diagram showing XopAI and HopU1 from *Pseudomonas syringae*. The structures are colored in the rainbow scheme. (**B**) Comparison of electrostatic surface of XopAI and HopU1. The regions of negative and positive potential are shown in red and blue, respectively; uncharged and hydrophobic surface areas are colorless. The central cleft on XopAI and the ADP-ribosylation site on HopU1 are marked with yellow ovals.

**Figure 3 ijms-20-05085-f003:**
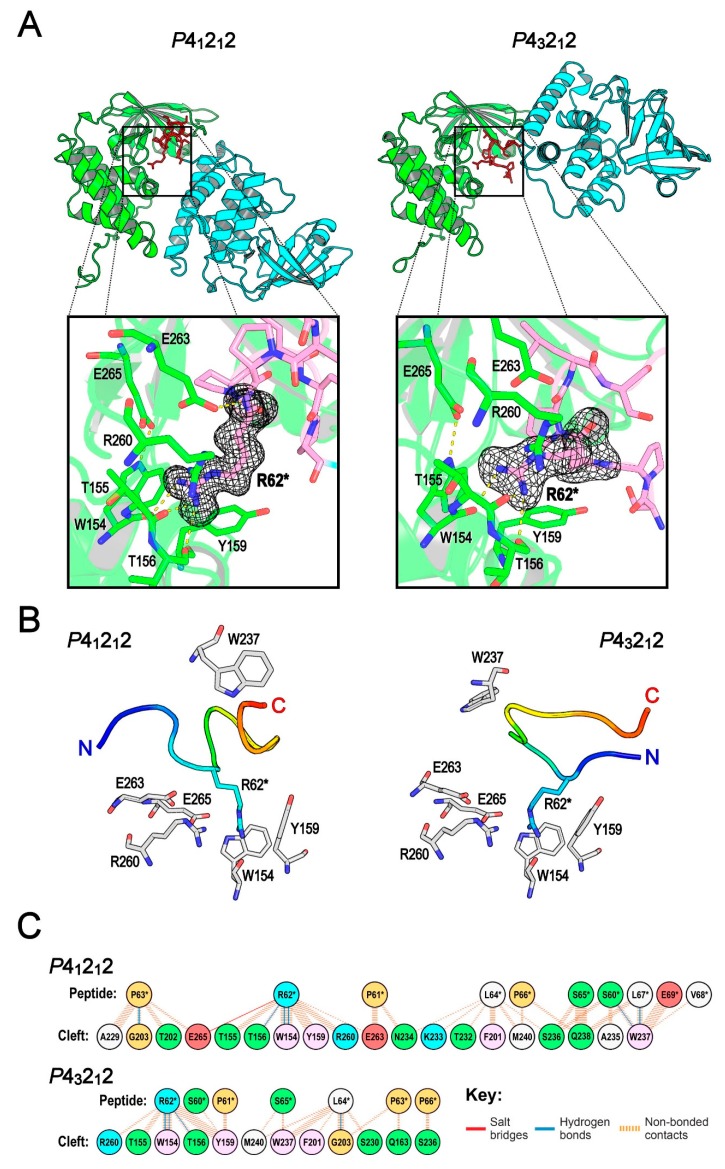
Two observed Arg peptide-binding modes of XopAI. (**A**) XopAI interaction found in the *P*4_1_2_1_2 and *P*4_3_2_1_2 crystals. The two interacting monomers are shown as ribbon models, and monomers A and B are colored in green and cyan, respectively. The bound N-terminal Arg peptide from monomer B is depicted as a red stick model. The insets show a zoomed-in view of the bound Arg (R62*). The carbon atoms of the bound Arg peptide are colored in pink; oxygen and nitrogen atoms are colored in red and blue, respectively. The black wire mesh depicts a *F*_o_–*F*_c_ electron density (contoured at 3 σ level) of R62*. The residues involved in R62* recognition are represented as stick models and labeled. Yellow dotted lines show potential hydrogen bonds or salt bridges. (**B**) The comparison of Arg peptide-binding modes found in the *P*4_1_2_1_2 and *P*4_3_2_1_2 crystals. For clarity, the bound Arg peptide is shown as a Cα trace and colored in the rainbow scheme; R62* is highlighted as a cyan stick model. W237 on the phosphate-nicotinamide (PN) loop and the residues involved in R62* binding are represented as gray stick models and labeled. (**C**) Residue interactions across the protein–peptide interface. The residues are colored to reflect their chemical properties (positive charged, cyan; negative charged, red; hydrophilic, green; aliphatic, white; aromatic, pink; proline or glycine, light orange). The number of cyan solid lines between any two residues indicates the number of potential hydrogen bonds between them. For non-bonded contacts, the width of the orange striped line is proportional to the number of atomic contacts between the two residues.

**Figure 4 ijms-20-05085-f004:**
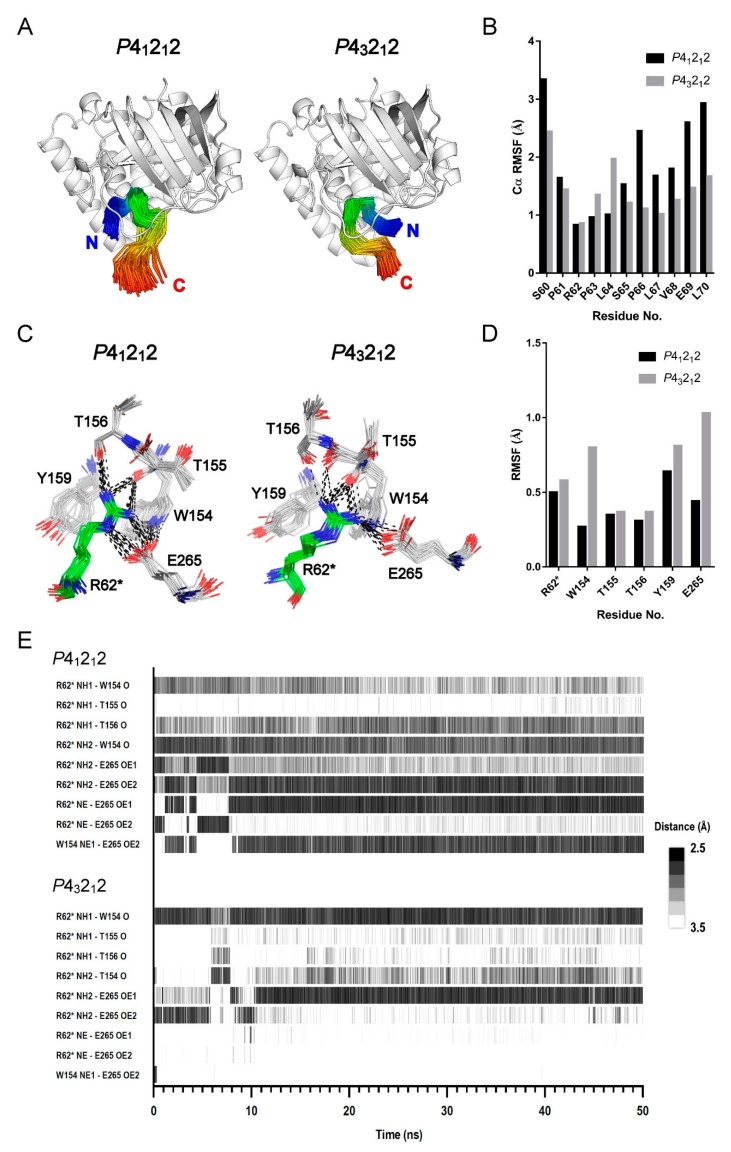
Molecular dynamics (MD) analysis of the two Arg peptide-binding modes. (**A**) Ensembles of Arg peptide conformation obtained from the last 25-ns MD simulation. The bound Arg peptide in every snapshot is shown as a Cα trace and colored in the rainbow scheme. (**B**) The root-mean-square fluctuation (RMSF) of the Cα atoms in the bound Arg peptide calculated from the results shown in (A). (**C**) Ensembles of Arg-binding poses during the last 25-ns MD simulation. The carbon atoms of the bound Arg (R62*) are shown in green, and those of the binding residues are in gray. The oxygen and nitrogen atoms are colored in red and blue, respectively. Potential hydrogen bonds or salt bridges are depicted as black dotted lines. (**D**) The RMSF of the residues in the R62*-binding site calculated from the results shown in (C). (**E**) The timeline of hydrogen bonds formed in the binding site during the 50-ns MD simulation. The hydrogen bonds formed between R62* and the binding residues are depicted in grayscale lines; the grayscale represents the hydrogen bond distance, which indicates the bond strength.

**Figure 5 ijms-20-05085-f005:**
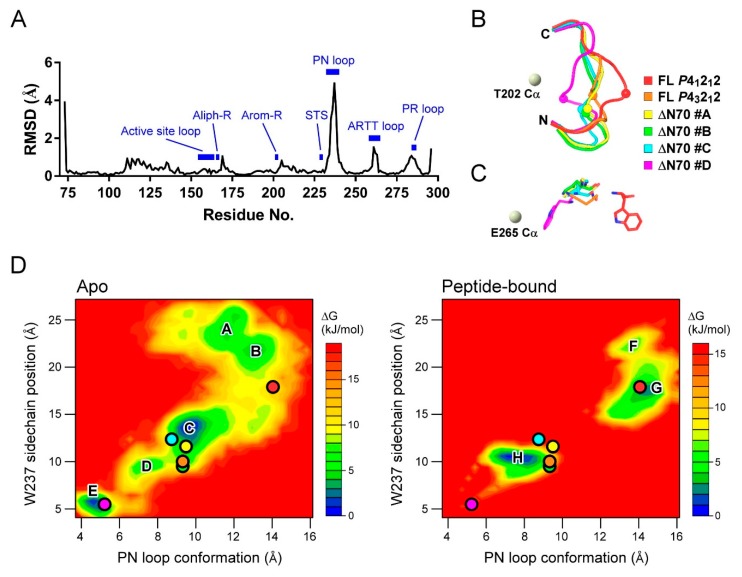
MD analysis of PN loop dynamics. (**A**) Per-residue averaged root-mean-square deviation (RMSD) among available XopAI crystal structures. Those regions of interest are marked as blues bars and labeled. (**B**) Observed PN loop conformations in the full-length (FL) and ΔN70 crystals. The position of W237 on the PN loop is shown as a bead model. As a reference, the location of Thr202 Cα atom is depicted as a gray bead. Protein backbones are colored distinctly for the various conformations as follows: FL *P*4_1_2_1_2 (red), FL *P*4_3_2_1_2 (orange), ΔN70 monomer A (yellow), ΔN70 monomer B (green), ΔN70 monomer C (cyan), and ΔN70 monomer D (magenta). (**C**) Observed W237 positions and conformations in the crystals; the view of this panel is rotated along the horizontal axis by 90° counterclockwise from that of (**B**). The carbon atoms of residue W237 are colored distinctly according to (**A**). The oxygen and nitrogen atoms are colored in red and blue, respectively. As a reference, the location of the E265 Cα atom is depicted as a gray bead. (**D**) energy landscapes depicting the motions of the PN loop and residue W237 in the absence (apo) and presence of Arg peptide (peptide-bound). The depth of the three-dimensional energy landscape indicates the value of the conformation free energy. The free energy is given in kJ/mol and is indicated by the color code shown in the figure. The conformation state of the observed structures is marked with colored circles. The color code for the circles is the same as that in (**B**,**C**).

**Table 1 ijms-20-05085-t001:** Data collection statistics for Br-MAD phasing.

Dataset	Native	Br-MAD		
		λ1 (H rem) ^1^	λ2 (infl)	λ3 (Peak)
Wavelength (Å)	0.9762	0.9056	0.9193	0.9190
Space group	*P*4_3_2_1_2			
Resolution (Å)	2.0	2.3	2.4	2.5
Redundancy	4.1	2.4	4.6	2.5
Completeness (%)	93.7 (85.3) ^2^	98.5 (99.7)	98.4 (99.7)	98.2 (99.7)
Average I/σ(I)	11.1 (2.5)	17.7 (3.5)	19.5 (4.0)	18.9 (3.8)
R_merge_ (%)	8.2 (52.5)	11.2 (49.2)	10.2 (44.7)	10.3 (43.6)

^1^ MAD data were collected at wavelengths corresponding to the high energy remote, inflection point, and peak of the atomic absorption edge. ^2^ Values in parenthesis indicate those for the highest resolution shell.

**Table 2 ijms-20-05085-t002:** Data collection and refinement statistics.

Protein	XopAI		XopAI-ΔN70
**Data collection**			
Wavelength (Å)	0.9762	0.9198	0.9762
Space group	*P*4_3_2_1_2	*P*4_1_2_1_2	*P*2_1_
Cell parameters (Å)	a = b = 73.05 Å, c = 114.06 Å	a = b = 52.98 Å, c = 212.09 Å	a = 62.78 Å, b = 98.76 Å, c = 77.45 Å, β = 91.21°
Resolution (Å)	2.01	1.53	2.26
Mosaicity (°)	1.07	0.29	1.23
Redundancy	4.1	10.9	3.5
Completeness (%)	93.7 (85.3) ^1^	100.0 (100.0)	99.4 (99.3)
Average I/σ(I)	11.1 (2.5)	9.6 (1.7)	7.5 (1.8)
R_merge_ (%)	8.2 (52.5)	13.1 (106.4)	14.2 (78.3)
CC_1/2_	0.997 (0.712)	0.997 (0.647)	0.990 (0.609)
**Refinement**			
Resolution limit (Å)	26.56–2.01	36.89–1.53	31.53–2.26
R_work_ (%)	16.6	16.1	18.9
R_free_ (%)	19.5	17.1	23.2
Number of non-H atoms			
Protein	1961	1973	7354
Water	236	374	558
Ramachandran plot statistics			
Favored regions (%)	99.15	98.73	98.43
Allowed regions (%)	0.85	1.27	1.46
Disallowed regions (%)	0	0	0.11
Average B factor (Å^2^)	25.9	22.7	35.1
R.m.s. deviation from ideality			
Bond length (Å)	0.004	0.010	0.004
Bond angle (˚)	0.655	1.024	0.561
Protein Data Bank (PDB) code	6KLY	6K93	6K94

^1^ Values in parenthesis indicate those for the highest resolution shell.

**Table 3 ijms-20-05085-t003:** The predicted binding energy between known peptide-binding domains and their peptides.

Domain Name	Binding Energy (kcal/mol)	Number of Bound Residues	Interface Area (Å^2^)	PDB ID
XopAI (*P*4_1_2_1_2)	−8.3	12	673	6K93 (this study)
XopAI (*P*4_3_2_1_2)	−7.6	7	470	6KLY (this study)
PDZ	−3.7 ± 2.3	4~9 (6) ^1^	432 ± 92	1BE9, 1L6O, 1MFG, 1N7F, 1OBX, 1OBZ, 1Q3P, 2I0I, 2QT5, 3DIW, 3LNY
IRS	−4.9 ± 1.3	9	617 ± 20	1UEF, 3ML4
VHS	−5.5 ± 3.2	5~7 (6)	420 ± 45	1JUQ, 1UJK
14-3-3	−5.7 ± 1.7	5~10 (7)	554 ± 122	1QJB, 2BTP, 2C74, 3MHR, 3UBW
WH1	−6.6 ± 2.6	5~6 (6)	315 ± 35	1DDV, 1EVH, 1QC6
Skp1	−6.9 ± 2.3	6~12 (10)	548 ± 151	2AST, 2OVQ, 2P1Q
SH2	−7.0	4	363	1FYR
TPR	−7.0 ± 1.1	5~8 (7)	502 ± 29	1ELR, 1ELW
PID	−7.7 ± 2.7	9~10 (10)	717 ± 28	1AQC, 1M7E, 1NTV
BIR	−8.3 ± 1.7	4~7 (6)	420 ± 54	1JD5, 1SE0, 3D9T
WD40	−8.4 ± 2.3	8~12 (9)	567 ± 44	2CE8, 4ERY, 5IGO, 5IGQ
SH3	−9.4 ± 0.6	9~10 (10)	480 ± 24	1CKA, 1N5Z, 1W70
SPRY	−10.0 ± 0.5	7	352 ± 2	2JK9, 3EMW

^1^ Values in parenthesis indicate the average number.

**Table 4 ijms-20-05085-t004:** Predicted binding free energies of XopAI-peptide complexes.

Complex	ΔE_vdW_ ^1^	ΔE_elec_	ΔG_polar_	ΔG_nonpolar_	ΔG_binding_
*P*4_1_2_1_2	−248.2 ± 25.2	−582.9 ± 42.2	496.4 ± 42.8	−33.8 ± 2.1	−368.5 ± 40.3
*P*4_3_2_1_2	−173.3 ± 22.1	−539.2 ± 40.4	505.1 ± 38.9	−27.7 ± 2.0	−235.1 ± 30.9

^1^ ΔE_vdW_, ΔE_elec_, ΔG_polar_, and ΔG_nonpolar_ are binding energy components of van der Waals, electrostatic, polar and nonpolar solvation energies, respectively. ΔG_binding_ is the total binding energy (ΔG_binding_ = ΔE_vdW_ + ΔE_elec_ + ΔG_polar_ + ΔG_nonpolar_). The unit of energy is kJ/mol. A positive value indicates the energy component is energetically unfavorable.

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
