# Peer review of "Crystal Structure-Based Exploration of Arginine-Containing Peptide Binding in the ADP-Ribosyltransferase Domain of the Type III Effector XopAI Protein"

_ijms, 2019, doi:10.3390/ijms20205085_

Round 1

Reviewer 1 Report

The manuscript by Liu et al is the revised paper of their previously submitted work. The author's have addressed all my concerns. 

Reviewer 2 Report

The authors answered to all raised concerns and improved the manuscript significantly. 

This manuscript is a resubmission of an earlier submission. The following is a list of the peer review reports and author responses from that submission.

Round 1

Reviewer 1 Report

Crystal structure-based exploration of arginine-containing peptide binding in the ADP-ribosyltransferase domain of the type III effector XopAI protein

The manuscript by Jyung-Hurng Liu and colleagues describes the crystal structure of XopAI mono-ADP-ribosyltransferase. The protein crystalized in two different space groups and in both structures the binding of arginine-containing peptide originating from the symmetry related molecule has been found. Authors speculate about the mechanism of peptide-binding on the basis of MD simulations, however, they do not provide any biochemical evidence (mutagenesis, binding etc.) for such interaction.  

Overall, this is interesting paper. It also contains novel findings that are worth publishing in the IJMS journal. Despite of the overall positive reception I would like to raise some major concerns.

MAJOR POINTS

1.     How was the resolution cut off made? The Rmerge values are relatively small and they should not be considered as any guidance for resolution cut off. Please report CC1/2 values. I/sigI values are high, thus one can imagine that your structures can be refined to better resolution. Please confront your data with the Karplus and Diedrisch reccomendations and possibly re-refine your structures or provide reasonable explanation why you cut the resolution at such values. (Science. 2012 May 25;336(6084):1030-3. doi: 10.1126/science.1218231.)

2.     Figure 3. The 2Fo-Fc map at 1sigma is not sufficient. Please provide bias-free Fo-Fc omit map countered at 3sigma

3.     Please discuss the lack of peptide binding studies and/or mutagenesis of crucial arginine residue. Is the protein monomeric or dimeric in solution?

4.     From the crystal structures and MD simulations is seems that the protein binds the arg residue in at least two different modes. Have you tried to probe the binding with the peptide alone in any binding assay using del70 variant? This would strengthen your conclusions significantly.

5.     Please provide validation reports from pdb deposition for all of the structures.

Reviewer 2 Report

In this manuscript Liu at el. determined the crystal structure of XopAI, a type III effector protein from Xac. Upon crystallization, the protein produced two types of crystals with different space groups. The authors found that XopAI shares sequence and structural similarities with proteins in the mono-ADP-ribosyltransferase family, however it has an altered active site and shows potential to bind arginine -containing peptide. Both the crystal structure and molecular dynamics simulation indicate that the protein recognizes arginine residue by hydrogen bonds and salt bride. 

The paper is well structure and clearly written, the necessary control experiments were presented. Therefore, the paper can be published in the presented form.